# GOLM1 and FAM49B: Potential Biomarkers in HNSCC Based on Bioinformatics and Immunohistochemical Analysis

**DOI:** 10.3390/ijms232315433

**Published:** 2022-12-06

**Authors:** Yue Xi, Tiange Zhang, Wei Sun, Ruobing Liang, Sridha Ganesh, Honglei Chen

**Affiliations:** 1Department of Pathology, School of Basic Medical Sciences, Wuhan University, Wuhan 430071, China; 2Department of Pathology, Zhongnan Hospital of Wuhan University, Wuhan 430071, China

**Keywords:** HNSCC, bioinformatics, GEO, TCGA, signaling pathways, biomarker, GOLM1, FAM49B

## Abstract

Head and neck squamous cell carcinoma (HNSCC) is one of the most common cancers worldwide. We aimed to identify potential genetic markers that could predict the prognosis of HNSCC. A total of 44 samples of GSE83519 from Gene Expression Omnibus (GEO) datasets and 546 samples of HNSCC from The Cancer Genome Atlas (TCGA) were adopted. The differently expressed genes (DEGs) of the samples were screened by GEO2R. We integrated the expression information of DEGs with clinical data from GES42743 using the weighted gene co-expression network analysis (WGCNA). A total of 17 hub genes were selected by the module membership (|MM| > 0.8), and the gene significance (|GS| > 0.3) was selected from the turquoise module. GOLM1 and FAM49B genes were chosen based on single-gene analysis results. Survival analysis showed that the higher expression of GOLM1 and FAM49B genes was correlated with a worse prognosis of HNSCC patients. Immunohistochemistry and multiplex immunofluorescence techniques verified that GOLM1 and FAM49B genes were highly expressed in HNSCC cells, and high expressions of GOLM1 were associated with the pathological grades of HNSCC. In conclusion, our study illustrated a new insight that GOLM1 and FAM49B genes might be used as potential biomarkers to determine the development of HNSCC, while GOLM1 and FAM49B have the possibility to be prognostic indicators for HNSCC.

## 1. Introduction

Head and neck squamous cell carcinoma (HNSCC), which mainly encompasses cancers of the lips, oral cavity, nasal cavity, paranasal sinuses, oropharynx, larynx, and nasopharynx, is the most common type of head and neck cancer. HNSCC accounts for more than 95% of the total incidence [1]. About 900,000 new cases of HNSCC occur each year, and more than 400,000 people die from it [2]. HNSCC occurs mainly in the mucosal epithelium of the pharynx and oral cavity. Currently, only physical examination can be relied on for HNSCC diagnosis, and there is no effective screening strategy [1]. Patients are usually given surgery, radiotherapy, chemotherapy, or treatment combining several kinds of intervening measures. However, there are still about 40~60% of treated patients who are unable to benefit from the treatment due to tumor local recurrence, metastasis to other parts of the body, and treatment resistance [3,4]. Therefore, investigating the potential mechanism of the occurrence and development of HNSCC and identifying specific molecular markers of HNSCC are conducive to the early diagnosis and treatment as well as prognosis analysis of HNSCC. It brings new possibilities for clinical treatment and ultimately improves the survival rate and quality of patients’ lives.

Recently, the continuous development of big data technology, and the improvement of computer and biological gene sequencing technology, have made it possible to analyze cancer-related genes at the molecular level. Bioinformatics, a science combining molecular biology and information technology, has been widely used in recent studies [5]. In addition to that, the continuous improvement of various medical research databases (GEO, ArrayExpress, Oncomine) and the large number of sequencing data being explored have triggered a global upsurge in the research on molecular evolution and gene function at the genome level. These convenient approaches create opportunities to access new cancer pathogenesis and potential therapeutic targets. Following high-throughput sequencing technology, which has increased, and gene chips that are widely used, bioinformatics has entered the stage of vigorous development to provide convenience for the study of diseases at the gene level. Many differential genes associated with the occurrence and development of tumors have been verified. The gene expression profiling chip has been widely applied to explore differential genes related to tumor diagnosis, prognosis, and treatment [6].

In this study, bioinformatics methods and techniques were adopted to analyze and integrate the mRNA expression data of HNSCC from GEO and TCGA. Differently expressed genes (DEGs) were screened by GEO2R and R software. The WGCNA was used to find the key modules associated with clinical traits, and hub genes were selected according to the correlation between the gene modules and clinical traits. Finally, Golgi membrane protein 1(GOLM1/GOLPH2/GP73) and CYFIP-related Rac1 interactor B (FAM49B) were selected out of the 17 hub genes. In addition, we conducted an immunohistochemistry analysis to identify the expression of GOLM1 and FAM49B in the tumor samples. Our study revealed that both FAM49B and GOLM1 expressed highly in HNSCC tissues and predicted a worse prognosis. 

## 2. Results

### 2.1. Identification of DEGs

Applying GEO2R and GEPIA2, 1876 DEGs of GSE83519 and2123 DEGs of the TCGA-generated RNA-seq data of HNSCC were eliminated. Finally, 293 DEGs related to HNSCC were obtained (Figure 1A).

### 2.2. WGCNA

The WGCNA was performed on 278 DEGs derived from the analysis of GSE83519 and TCGA databases to find the critical modules associated with clinical traits. Clinical traits such as tumor stages, clinical stages, pathologic stages, and living statuses were also retrieved from the GSE42743. By setting the soft-thresholding power at five (scale-free R2 = 0.87, Figure 1B,C) and the cut height at 0.25, six modules (non-clustering DEGs sorted into the grey module, Figure 1D) were identified. The dendrogram and heatmap of the genes indicated no apparent interactions among different modules. Therefore, the conclusion could be drawn that these modules had a high degree of independence (Figure 1E). As the heatmap of the module–traits correlations indicated, the turquoise module was most highly correlated with clinical traits, especially in the tumor stage (module–trait correlation = −0.39, *p* = 5 × 10^−4^ < 0.01, Figure 1F). The turquoise module consisted of 100 genes in total.

### 2.3. KEGG and GO

To analyze the biological functions and KEGG pathways of turquoise modules, DAVID accomplished GO and KEGG analyses. The top ten KEGG and GO results that were significantly enriched by *p*-value screening are shown in Figure 2. Gene ontology analysis indicated that the genes in turquoise modules were particularly enriched in an extracellular matrix organization, extracellular matrix structural constituent, extracellular region, integrin binding, and extracellular matrix. The KEGG pathway enrichment analysis indicated that proteoglycans in cancer, human papillomavirus infection, focal adhesion, pathways in cancer, and protein digestion and absorption were the most enriched pathways.

### 2.4. Selection of Hub Genes

We selected 17 hub genes from the turquoise module based on the module membership (|MM| > 0.8) and the gene significance (|GS| > 0.3) (Figure 2C). The hub genes were listed as follows: CYP4F22, HOXC9, PXDN, PARVB, CD207, FADD, FBLIM1, CYP2C18, SHOX2, COL1A2, ACPP, COL3A1, MAD2L1, CENPK, FAM49B, GOLM1, and CEACAM6. A heatmap of the 17 hub genes was constructed according to their topological overlap matrix. According to the heatmap, there were strong associations among all these hub genes except for PARVB (Figure 2D). Based on the results of survival analysis, GSEA and TIMER2.0, we selected two genes, namely GOLM1 and FAM49B, which we were most interested in for subsequent analysis (the analysis results of all 17 genes are attached to the Appendix A).

### 2.5. Prognostic Value of GOLM1 and FAM49B in HNSCC

To estimate the prognostic value of GOLM1 and FAM49B in HNSCC, we generated a Kaplan–Meier curve based on the TCGA database. The optimal cutoff value was obtained by ROC analysis between the patients’ expression and survival time. Based on the cutoff value of the expression, the patients were divided into high-expression and low-expression groups. The high GOLM1 expression group (*p* = 0.05, Figure 3A) and high FAM49B expression group (*p* = 0.039, Figure 3B) were significantly correlated with worse 10-year overall survival, which indicated the predictive value of high GOLM1 expression and high FAM49B expression for lower survival in HNSCC patients.

### 2.6. PPI Analysis of GOLM1 and FAM49B

According to the ranks of the predicted functional partners, it was clear that GOLM1 is mainly associated with Clusterin (CLU) (0.915), Golgi phosphoprotein 3 (GOLPH3) (0.820), and the epidermal growth factor receptor (EGFR) (0.793). While FAM49B did not show much correlation with the other genes, whose highest score of predicted functional partners was 0.680 (Figure 3C,D), the combined score indicated the correlation between the genes and proteins (Appendix A). The higher the score was, the greater the correlation was. 

### 2.7. Correlation Analyses between GOLM1 and FAM49B Expression and Immune Cell Infiltration

We used the gene module in the TIMER 2.0 database to perform the correlation analyses between the GOLM1 and FAM49B expression and immune cell infiltration in HNSCC (Appendix A). Using XCELL as the algorithm, it was shown that GOLM1 expression was positively correlated with the infiltrating levels of the T cell CD8^+^ effector memory, T cell CD8^+^ central memory, T cell CD4^+^ naive, T cell CD4^+^ memory, T cell CD4^+^ Th2, regulatory T cells (Tregs), macrophages (M1 and M2), endothelial cell, myeloid dendritic cell, and myeloid dendritic cell activated. Simultaneously, it was negatively correlated with T cell CD4^+^ central memory, B cell plasma, and neutrophil (*p* < 0.05). Through the figure below, most of the immune cells mentioned were positively associated with GOLM1 levels, which demonstrated that HNSCC patients with a high expression of GOLM1 might have more infiltrating immune cells in the tumor tissues.

When it came to FAM49B, Appendix A shows that FAM49B expression was positively correlated to T cell CD4^+^ memory, T cell CD4^+^ Th2, B cell naïve, and Macrophage, while also appearing negatively related to T cell CD4^+^ central memory, T cell CD4^+^ Th1, B cell, B cell memory, and B cell plasma. Comparing the correlation level between the two genes, FAM49B had a lower correlation level with immune infiltration cells than GOLM1.

### 2.8. GSEA

We further studied the mechanism of the GOLM1 and FAM49B on the prognosis of HNSCC using GSEA v4.10 based on TCGA datasets. The results proved that GOLM1 and FAM49B were significantly enriched in some pathways (Table 1).

When GOLM1 was overexpressed, 36 gene sets were differentially enriched. The top ten intersected gene-enriched pathways associated with GOLM1 upregulation included cell adhesion molecules (CAMs), complement and coagulation cascades, the TGF-βsignaling pathway, focal adhesion, vascular smooth muscle contraction, ECM–receptor interaction, colorectal cancer, systemic lupus erythematosus, pathways in cancer, and Nglycan biosynthesis (Appendix A). When FAM49B was overexpressed, 41 gene sets were differentially enriched. The top ten intersected gene-enriched pathways associated with FAM49B upregulation were antigen processing and presentation, oocyte meiosis, RNA degradation, NOD-like receptor signaling pathway, cytosolic DNA-sensing pathway, ubiquitin-mediated proteolysis, RIG-I-like receptor signaling pathway, basal transcription factors, proteasome, and pyrimidine metabolism (Appendix A).

It is worth noting that some enriched pathways of GOLM1 and FAM49B are related to immune surveillance, such as the Wnt and chemokine signaling pathways of GOLM1, the p53 signaling pathway, and the Jak-STAT signaling pathway of FAM49B. These two genes may participate in the inhibition of tumor progression and have a certain reference value for finding new immune checkpoints.

### 2.9. Expression of GOLM1 and FAM49B in HNSCC Tissues

As the immunohistochemistry (IHC) results indicated, the GOLM1 protein was found to be expressed in most HNSCC. The GOLM1 protein was localized in the cytoplasm of tumor cells (Figure 4), with higher levels in poorly differentiated HNSCC tissues. The GOLM1 protein was negative in the normal squamous epithelium (Figure 4A) but positive in the normal glandular epithelial cells (Figure 4B) and fibroblast cells. The FAM49B protein was mainly detected in the cytoplasm, the nuclear of tumor cells and immune cells, but a negative signal was in the normal squamous epithelium (Figure 5A–F). Interestingly, the positive signal of GOLM1 protein expression was a punctiform staining pattern, while the FAM49B showed the patched signal. Cytokeratin (CK) can label the tumor cells of HNSCC and normal epithelial cells, which can discriminate from the immune cells of the tumor microenvironment, such as CD3 labeling T lymphocytes, CD20 meaning B lymphocytes, and CD68 labeling macrophages. These interesting results demonstrated that the co-expression of GOLM1 and CK, GOLM1 and FAM49B, GOLM1 and CD3, FAM49B and CD3, GOLM1 and CD68, and FAM49B and CD20 were observed in the multiplex immunofluorescence (mIF) results (Figure 6A–P).

### 2.10. Clinicopathological Significance of GOLM1 and FAM49B Expression in HNSCC

The association between the GOLM1 and FAM49B expression and clinicopathological characteristics in HNSCC tissues was investigated. A total of 81 valid results from the TMAs were selected to perform the Chi-squared analysis. The results of IHC were evaluated. We defined the levels ≤2 of GOLM1 expression as low expression while the levels >2 were defined as high expression. As for the expression of FAM49B, we used absolute positive (score = 1) and negative (score = 0) scores to sort the results.

The correlation between the expression of GOLM1 and FAM49B in tumor cells and clinicopathological characteristics was first studied. Then, we performed the Chi-squared analysis to find whether there was a potential linkage between the levels of GOLM1 expression in the tumor cells and clinicopathological characteristics. As listed in Appendix A, our results show that GOLM1 alone in the tumor cells was associated with the pathological grade in HNSCC (*p* = 0.004). While no significant correlations were found between FAM49B expression in the tumor cells and clinicopathological factors.

As Table 2 indicates, a high expression of the GOLM1 protein was significantly associated with a higher pathological grade (*p* < 0.01) in HNSCC, though more evidence should be taken into account.

### 2.11. Correlation between GOLM1 and FAM49B Expression in Tumor Cells

HNSCC tumor cores from TMAs were selected as cases and divided into four groups according to the expression of GOLM1 and FAM49B. Eighty-one valid results from the TMAs were selected. From the results of SPSS, we found that the continuity correlation *p*-value was 0.518 (>0.05), which meant there was no definite correlation between the expression of GOLM1 and FAM49B in HNSCC.

## 3. Discussion

The incidence of HNSCC is still increasing year by year [1]. Due to the limited improvement in the patient’s prognosis with traditional treatments, the epidermal growth factor receptor, tyrosine kinase inhibitors, and immune checkpoint inhibitors represented by cetuximab undoubtedly provide significant progress in the treatment of HNSCC when it comes to the clinical application [7,8]. Early diagnosis and therapy can improve the prognosis of patients, but there is still a lack of effective biomarkers for cancer screening and the prognosis of HNSCC. Therefore, it is necessary to find valuable biomarkers in HNSCC.

WGCNA has been widely applied in various biological contexts and has attained significant progress. Using the module eigengene, WGCNA can identify different clusters (modules) of genes with similarities in expression. In addition, it explores the correlation between modules and clinical traits, making a tremendous difference in the selection of hub genes. However, in our study, the amount of our 273 DEGs may decrease the stability of the WGCNA network compared to using all genes without the criterion we chose.

In this study, 44 samples of GSE83519 from GEO datasets and 546 samples of HNSCC from TCGA were adopted, and DEGs were screened by GEO2R and R software. As a result, 293 DEGs related to HNSCC were obtained. RNA-seq data and clinical information of GSE42743 were applied. After performing WGCNA analysis, the hub genes were selected based on the correlation between the gene modules and clinical traits, according to the criteria of |MM|>0.8 and |GS|>0.3. Four different analyses were utilized to screen our interesting genes: GOLM1 and FAM49B. We conducted single-gene analyses to determine their potential clinical meanings using TIMER2.0 and GSEA. Finally, we performed IHC and mIF to explore the correlation between protein expression and clinicopathological characteristics in HNSCC tissues.

FAM49B has been confirmed to modulate cellular actin assembly through the RAC-Pak Axis, drive cytoskeletal remodeling, and regulate pseudopod complexity [9,10,11]. Simultaneously, FAM49B can inhibit TCR signal transduction and negatively manage T cell activation through modulating cytoskeletal remodeling [11]. The overexpression of FAM49B has been identified in pancreatic ductal adenocarcinoma (PDAC), early pancreatic intraepithelial neoplasia (PanIN), and breast cancer. Furthermore, high expression of FAM49B can predict a worse prognosis [12,13,14], which was consistent with the results of our study in HNSCC. We found positive staining for the FAM49B protein in the tumor cells of HNSCC, and the co-expression of FAM49B and CD3, CD20, and CD68 demonstrates that FAM49B can express in the T cells, B cells, and macrophages around SCC.

Previous studies have found different mechanisms of tumor invasion and proliferation mediated by FAM49B in different cancers. Taspase 1 (TASP1) promotes the proliferation and migration of gallbladder cancer cells by targeting the upregulation of FAM49B through the TASP1-PI3K/Akt-FAM49B axis [12]. FAM49B can also stimulate breast cancer cell proliferation and migration through the Rab10/TLR4 pathway [13]. In colorectal cancer (CRC) and hepatocellular carcinoma (HCC), FAM49B, as a downstream target of the Zinc Finger RNA binding protein (ZFR), may be a potential tumor suppressor [15]. However, the study of FAM49B on the invasiveness of HNSCC cells is still blank, and the specific mechanism of FAM49B mediating in HNSCC remains to be further explored and demonstrated.

GOLM1 is a membrane protein located on the Golgi apparatus, which is mainly expressed by cells of the epithelial lineage and upregulated by virus infection [16]. Our study found that compared with normal tissues, the expression of GOLM1 was increased in tumor tissues, and the degree of increase may be correlated with the differentiation degree of the tumor. Early studies have shown that increased GOLM1 expression in hepatocytes seems to be a general feature of advanced liver disease [17]. Simultaneously, the high expression of GOLM1 is associated with a variety of cancers, including prostate cancer, non-small cell lung carcinoma (NSCLC), and HCC, and GOLM1 has been considered an early diagnostic marker of HCC [18,19,20,21]. In HCC, GOLM1 upregulates the expression of PD-L1 in cancer cells through the deubiquitination of PD-L1, which is mediated by the enzyme deubiquitination, thereby benefiting cancer cells to escape from immune cells [22]. It has also been shown that deoxycholic acid (DCA) can upregulate the expression and release of GOLM1 by activating the nuclear factor-Kappa B (NF-κB) pathway and destroying the Golgi apparatus in chronic liver disease (CLD) and HCC [22]. Thus, GOLM1, a multifunctional protein, plays an essential role in facilitating cancer cells’ epithelial-mesenchymal transition (EMT) and inducing cancer metastasis [23]. However, according to the current research results, GOLM1 may be a kind of housekeeping gene according to its expression in most normal tissues. In past studies, the intracellular and extracellular interaction between GOLM1 and secretary clusterin (sCLU) has been found. The intracellular colocalization of GOLPH2 and sCLU in Golgi has also been confirmed. According to the role of each protein in this protein–protein association, it is speculated that GOLM1 might participate in the sCLU post-translational modification, transportation, and secretion [24]. Applying GSEA analysis, we discovered that GOLM1 was markedly enriched in the TGF-β signaling pathway and Wnt signaling pathway in HNSCC. Some previous researchers have elaborated on the connection between GOLM1 and the TGF-β1/Smad2 signaling pathway in bladder cancer and HCC [25,26]. GOLM1’s effect via the Wnt/β-catenin signaling pathway has also been shown in human glioblastoma, which can promote its proliferation and motility [27]. Our study was the first to discover the potential function of GOLM1 with the TGF-β signaling pathway and Wnt signaling pathway in HNSCC. The specific biological function and related molecular mechanism of GOLM1 in HNSCC have not been determined.

What is more, by utilizing TIMER2.0, our study identified that GOLM1 was correlated to T cell CD8+ effector memory, T cell CD8^+^ central memory, T cell CD4^+^ naive, T cell CD4^+^ memory, T cell CD4^+^ Th2, Tregs, macrophages (M1 and M2), myeloid dendritic cells, myeloid dendritic cell activated, T cell CD4^+^ central memory, B cell plasma, and Neutrophils. Moreover, GOLM1 has a close correlation with cancer immunosuppression, which can promote PD-L1 stabilization, and with the transportation of PD-L1 into the tumor-associated macrophages with exosome dependence [28]. On the other hand, FAM49B was associated with CD4^+^ T cells, B cells, CD8^+^ T cells, macrophages, neutrophils, and dendritic cells. Immune-related mechanisms in HNSCC and immunization strategies may provide a potential direction to the diagnosis, treatment, and prognosis. TIMER2.0 indicated that GOLM1 and FAM49B had something to do with the regulation of immune infiltrating cells in HNSCC. Our mIF results also confirmed the co-expression of GOLM1 and FAM49B with T cells, B cells, and macrophages. However, we need further analysis and experiments to determine how the two genes functionally work and how many pathways participate in the regulation.

Our study is the first to systematically report that the hub genes GOLM1 and FAM49B are associated with the prognostic value of HNSCC throughout the process of data analysis and experimental verification. These findings demonstrated that GOLM1 and FAM49B genes might be used as potential biomarkers to determine the development of HNSCC; moreover, GOLM1 and FAM49B can possibly be verified as prognostic indicators of HNSCC patients.

However, limited by the finite data of HNSCC cases and experiments, our findings may not be applied to all HNSCC patients. Differences in different geographical regions and pathogenic factors can result in diverse prognostic biomarkers of HNSCC, on which we will further focus. As for now, the specific biological function and related molecular mechanism of GOLM1 and FAM49B remain undiscovered in HNSCC. Further research is essential to reveal the function of GOLM1 and FAM49B in HNSCC cells, explore the interaction among GOLM1, FAM49B, and other genes, and build mouse tumor-bearing models to find potential drugs for HNSCC treatment.

## 4. Materials and Methods

### 4.1. Microarray Datasets

GENE EXPRESSION OMNIBUS (http://www.ncbi.nlm.nih.gov/geo/ accessed on 21 January 2021), commonly referred to as GEO, is a created gene expression database retained by the National Center for Biotechnology Information (NCBI) in the United States. Studies from the GEO database were considered eligible and satisfactory based on the following criteria: (1) studies with HNSCC tissue samples, (2) studies including adjacent normal tissues as the control, (3) all HNSCC tissues and normal tissues confirmed by histopathology and (4) datasets containing more than 20 pairs of HNSCC tissues and normal tissues. According to the above criteria, we downloaded and collected 22 normal mucosal tissues and 22 HNSCC tissues through a high-throughput gene expression dataset numbered GSE83519. The RNA-seq data and clinical information of GSE42743 were applied.

The Cancer Genome Atlas (TCGA) project is a multi-center institutional effort supported by the National Institute of Health to provide a comprehensive genetic analysis of different cancers and establish correlations with clinical outcomes [29]. We downloaded the RNA sequencing datasets (TCGA-HNSC) of 502 HNSCC tissues and 44 adjacent normal tissues, including clinical information and gene expression from TCGA. Incomplete data were deleted before analysis.

### 4.2. Identification of DEGs

GEO2R conducts comparisons on original submitter-supplied processed data tables utilizing the GEOquery and limma R packages from the Bioconductor project (http://www.ncbi.nlm.nih.gov/geo/geo2r, accessed on 21 January 2021). Using GEO2R, we compared and analyzed the 22 HNSCC tissues with 22 corresponding normal tissues. Fold-change (FC) and *p*-values were the standard parameters to screen differentially expressed genes (DEGs), which were set as the criteria of |log(FC)| > 1, *p*-value < 0.01.

Gene expression profiling interactive analysis (GEPIA2) provides customizable functions such as tumor/normal differential expression analysis, profiling according to cancer types or pathological stages, and patient survival analysis [30]. It was applied to identify the DEGs of 502 HNSCC tissues and 44 adjacent normal tissues from TCGA according to the criteria of |log(FC)| > 1, adjusted *p*-value < 0.01.

### 4.3. Weighted Gene Co-Expression Network Analysis

The weighted gene co-expression network analysis (WGCNA) is a systematic biological method. WGCNA analyzes the interaction patterns among genes and separates genes into various modules. It also plays a significant role in exploring the correlation between gene modules and clinical traits. The probe information of the 15 DEGs we selected was not found in the samples in GSE42743. Therefore, we selected the expression of the other 278 DEGs and clinical traits (tumor stages, etc.) from 74 oral cavity cancer samples in GSE42743 to perform WGCNA analysis using the R package WGCNA [31].

Hub genes were selected based on the correlation between the gene modules and clinical traits. Module membership (MM) and gene significance (GS) were also taken into consideration.

### 4.4. GO and KEGG Pathway Analysis

Gene ontology (GO) is specifically designed to support the computational representation of biological systems, providing a framework to describe the functions of gene products from all organisms [32]. KEGG (http://www.kegg.jp/ or http://www.genome.jp/kegg/, accessed on 24 February 2022 ) is an encyclopedia of genes and genomes aimed at assigning functional meanings to genes and genomes at both the molecular and higher levels [33]. The Database for Annotation, Visualization, and Integrated Discovery (DAVID) 6.8 (https://david.ncifcrf.gov, accessed on 24 February 2022) provides a comprehensive set of functional annotation tools that can be applied to gene functional enrichment analysis. The Ggplot2R package was used to visualize GO terms and KEGG pathways information by implementing high-quality figures [34].

### 4.5. Survival Analysis of Hub Genes

Receiver operating characteristic (ROC) curves were plotted with SPSS 22.0 software to assess the optimal cutoff value for survival analyses. To evaluate the prognostic value of the screened candidate genes, the overall survival rate (OS) was determined by the Kaplan–Meier curve with the log-rank test and was conducted and plotted by the survival R package. All clinical information used for survival analysis was derived from TCGA.

### 4.6. PPI Analysis of Two Single Genes

The interaction between proteins is crucial to find out the metabolic and molecular mechanisms of tumors. The String database (http://string-db.org, accessed on 27 February 2022) provides a critical assessment of protein interactions. The single-gene analysis of GOLM1 and FAM49B, which were selected from the hub genes, was performed on the website.

### 4.7. GSEA

To identify the intersected genes enriched pathways, we employed Gene Set Enrichment Analysis (GSEA, http://software.broadinstitute.org/gsea/index.jsp, accessed on 10 February 2022) to run KEGG gene sets of GOLM1 and FAM49B [35]. The gene expression data of 502 samples of HNSCC extracted from TCGA were divided into the upper and the lower group according to the levels of gene expression. Additionally, “c2.cp.kegg.v7.5.symbols.gmt gene sets” of the MSigDB collection were chosen as the gene set database, with the following parameters: *p*-value cutoff  = 0.05, false discovery rate (FDR) = 0.25, |NES| > 1.5, and gene size ≥ 25. The number of permutations was set at 1000 during the Gene Set Enrichment Analysis process.

### 4.8. TIMER2.0 Database Analysis

The correlations between hub genes and infiltrated immune cells were explored in the TIMER2.0 database (http://timer.cistrome.org/, accessed on 20 October 2021). It has access to visualized functions and comprehensively analyzes tumors infiltrating into immune cells, including B cells, CD4^+^ T cells, CD8^+^ T cells, neutrophils, macrophages, and dendritic cells (DCs). Instead of just utilizing one algorithm, TIMER2.0 provided a more robust estimation of immune infiltration levels for The Cancer Genome Atlas (TCGA) or user-provided tumor profiles using six state-of-the-art algorithms [36].

### 4.9. Tissue Microarrays

Two separate HNSCC tissue microarrays (TMAs) were purchased and used in this study (Guilin Fanpu Biotechnology Co., Guilin, China). The TMA slides consisted of 83 HNSCC tissues and 5 noncancerous tissues. All 88 cores with a 1.5 mm diameter were arranged in paraffin blocks in TMAs. Detailed clinicopathological features, such as gender, age, tumor grades, depth of tumor invasion (T), lymph node metastasis (N), and distant metastasis (M), are given in Appendix A.

All procedures involving human participants were in accordance with the ethical standards of the Guilin Fanpu Biotechnology Co. All research was in compliance with the terms of the 1964 Declaration of Helsinki and its later amendments or comparable ethical standards. Because of this type of retrospective study, informed consent was not required.

### 4.10. Immunohistochemistry Analysis

IHC analysis was conducted to detect GOLM1 and FAM49B expression in HNSCC tissues. TMA sections were firstly deparaffinized and rehydrated. The antigen retrievals of GOLM1 and FAM49B were applied in citrate acid buffer (10 mM, pH 6.0) for 15 min by a microwave. The sections were incubated with a rabbit anti-human GOLM1 polyclonal antibody (1:400 dilution, NBP1-88775, Novus Biologicals, Centennial, CO, USA) and rabbit anti-human FAM49B polyclonal antibody (1:50 dilution, NBP1-88582, Novus Biologicals, USA) at 4 °C overnight. Subsequently, horseradish peroxidase (HRP) conjugated to the goat anti-mouse/rabbit secondary antibody was added to the slides at 37 °C for 30 min of incubation. 3,3′-diaminobenzidine (DAB) chromogen (DAKO, Santa Clara, CA, USA) and nuclear counterstaining with hematoxylin were executed afterward. Human stomach tissue for GOLM1 and human lymph node for FAM49B were regarded as the positive control. Primary antibodies replaced by PBS were regarded as the negative control.

### 4.11. Evaluation of Immunohistochemistry

Immunostaining reactivity was observed using light microscopy (Olympus BX-53 with CCD DP74, Shinjuku, Japan). The results were analyzed by two pathologists (Xi Y and Chen H) who acted independently and were blind to the clinicopathological characteristics of the study. The analyses of the two pathologists were compared, and any discrepancies were reassessed to arrive at a consensus. The expressions of GOLM1 and FAM49B in tumor and immune cells were recorded for further research.

### 4.12. Evaluating Results of GOLM1 and FAM49B Expression

The GOLM1 expression in the tumor cells was graded as four levels ranging from 0 to 3. The levels were determined by the ratio of positive cells to the total cells in the percentage as follows: 0 (0%), 1 (0–10%), 2 (10–50%), and 3 (>50%). On the other hand, FAM49B expression in tumor cells was graded in absolute figures, with 0 in the case of negative expression and 1 for positive expression.

### 4.13. Multiplex Immunofluorescence Staining

We accomplished manual mIF staining using the Opal 7-Color IHC Kit (Akoya, Waltham, MA, USA) in a section obtained from Formalin-fixed paraffin-embedding (FFPE) HNSCC tissues. We used a Vectra 3.0 multispectral imaging system (Akoya, Waltham, MA, USA) to scan the stained slides.

The immunofluorescence markers consisted of GOLM1 (1:800 dilution, NBP1-88775, Novus Biologicals, USA), FAM49B (1:100 dilution, NBP1-88582, Novus Biologicals, USA), CK (AE1/AE3), CD3 (F7.2.38), CD20 (L26), and CD68 (PG-M1). The last four were ready-to-use antibodies from Agilent/DAKO, Santa Clara, CA, USA.

The antigen retrievals were performed before attaching each primary antibody to the section using a Meidi microwave (M1-L213C, Meidi, Beijiao, China). The primary antibodies were incubated and visualized using tyramide signal amplification linked to specific fluorochrome from the mIF Kit for each primary antibody. The whole mIF procedure was executed according to the manufacturer’s instructions. To create the spectral library, the uniplex IF was used with each antibody. This multispectral analysis also adopted the same fluorochrome used in the mIF in human FFPE HNSCC tissues. HNSCC tissues were also analyzed in the same mIF procedure, respectively, with or without the primary antibodies, to establish the positive and negative (autofluorescence) controls.

A Vectra 3.0 multispectral microscope system and InForm2.6 software were used to scan the mIF and uniplex IF-stained slides under fluorescent illumination. From each slide, the Vectra automatically captured the fluorescent spectra from 420 nm to 720 nm at 20 nm intervals with the most appropriate exposure time. Next, the captured images were combined to create a single stack image that retained the particulate spectral signature of all the IF markers.

### 4.14. Statistical Analysis

Statistical analyses were fulfilled with R software (Version 4.1.2) and SPSS 22.0 software (Chicago, IL, USA). The correlation between GOLM1 and FAM49B expression in the tumor cells and the potential relationship between their expression and the clinicopathological parameters of the HNSCC tissues were explored by the χ2 test. *p* < 0.05 was considered statistically significant in all analyses.

## 5. Conclusions

In conclusion, our experiment screened out GOLM1 and FAM49B in relation to the development of HNSCC through the bioinformatics method. We indicated that the high expression of both FAM49B and GOLM1 in HNSCC tissues predicted a worse prognosis, possibly as a result of the regulation of immune infiltrating cells in the tumor environment. IHC and mIF revealed special spatial expression patterns of GOLM1 and FAM49B in the HNSCC; moreover, a high expression of GOLM1 might be associated with a higher pathological grade. These results improve our understanding of the differential expression of GOLM1 and FAM49B in HNSCC and also indicate that GOLM1 and FAM49B may be used as bioinformatics markers for the assessment of HNSCC prognosis and as potential targets for HNSCC treatment, which is conducive to the development of personalized immunotherapy for HNSCC in the future.

## Figures and Tables

**Figure 1 ijms-23-15433-f001:**
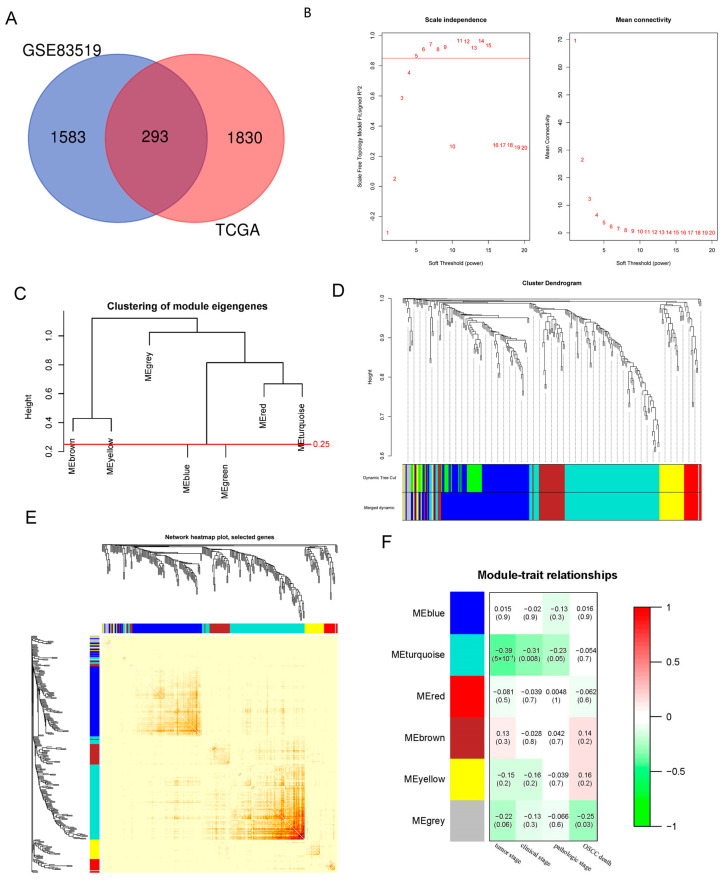
Identification of key modules correlated with clinical traits through WGCNA. (**A**)Venn diagram shows the intersecting DEGs from GEO and TCGA. A total of 293 DEGs are shared by GSE83519 (blue) and TCGA-HNSC (red). (**B**) Analysis of the scale independence (**left**) and the mean connectivity (**right**) for different soft-thresholding powers. Five was set as the soft-thresholding power, meanwhile the scale free R2 was 0.87 > 0.85. (**C**) Clustering of module eigengenes. The cut height (0.25) was indicated by the red line. (**D**) Dendrogram of 278 DEGs clustered based on a dissimilarity measure (1-TOM), each module containing 10 genes at least. The merged dynamic contained six modules after combining the green and blue modules which were under the cut height (0.25). (**E**) Dendrogram and heatmap of DEGs. The color intensity varied positively with the correlation of different DEGs according to the topological overlap matrix (TOM). (**F**) Heatmap of the correlation between module eigengenes and clinical traits. The *p* value and module–trait correlation were contained in each cell.

**Figure 2 ijms-23-15433-f002:**
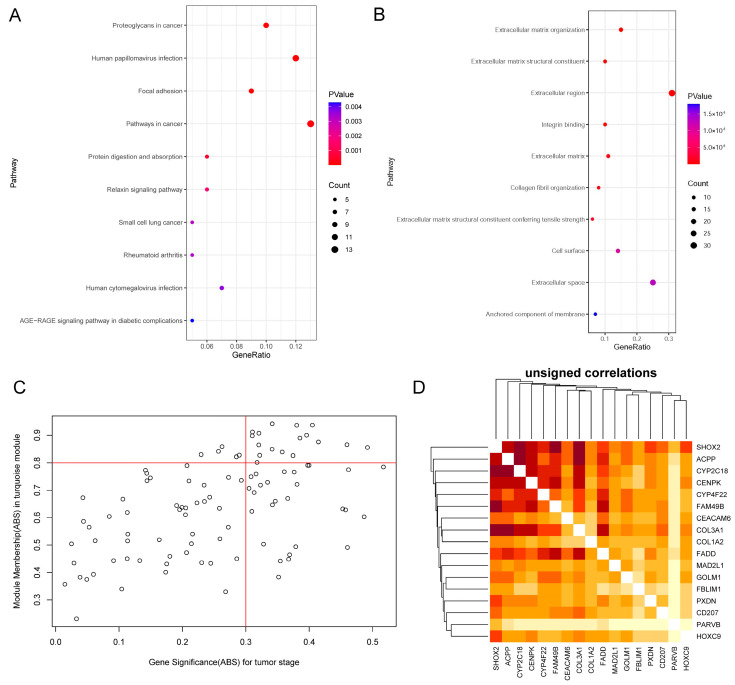
Functional annotation of key modules and selection of hub genes. GO categories and KEGG pathways annotation of the DEGs in HNSCC. Gene count is represented by dot size. *p*-value is represented by color. (**A**) GO categories. (**B**) KEGG pathways. (**C**) The scatter plot of module eigengenes in the turquoise module. The two red lines indicate the standards of the selection of the hub genes (module membership (ABS) > 0.8 and gene significance (ABS) > 0.3). Each hollow circle represents one gene in the turquoise module. (**D**) The heatmap of 17 selected hub genes is based on their topological overlap matrix (TOM). Color intensity varies positively with the correlation of hub genes.

**Figure 3 ijms-23-15433-f003:**
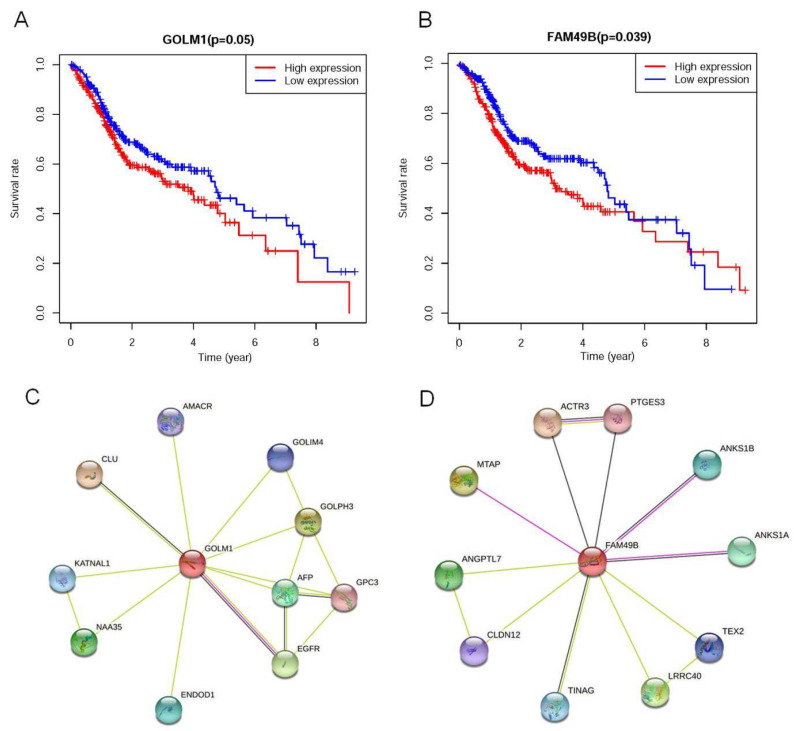
The effect of different gene expressions on OS in HNSCC and PPI network of GOLM1 and FAM49B. (**A**) K–M analysis of OS between the high FAM49B expression group and low FAM49B expression group in HNSCC. (**B**) K–M analysis of OS between the high GOLM1 expression group and low GOLM1 expression group in HNSCC. (**C**) Genes correlated with GOLM1 were shown. (**D**) Genes correlated with FAM40B are shown.

**Figure 4 ijms-23-15433-f004:**
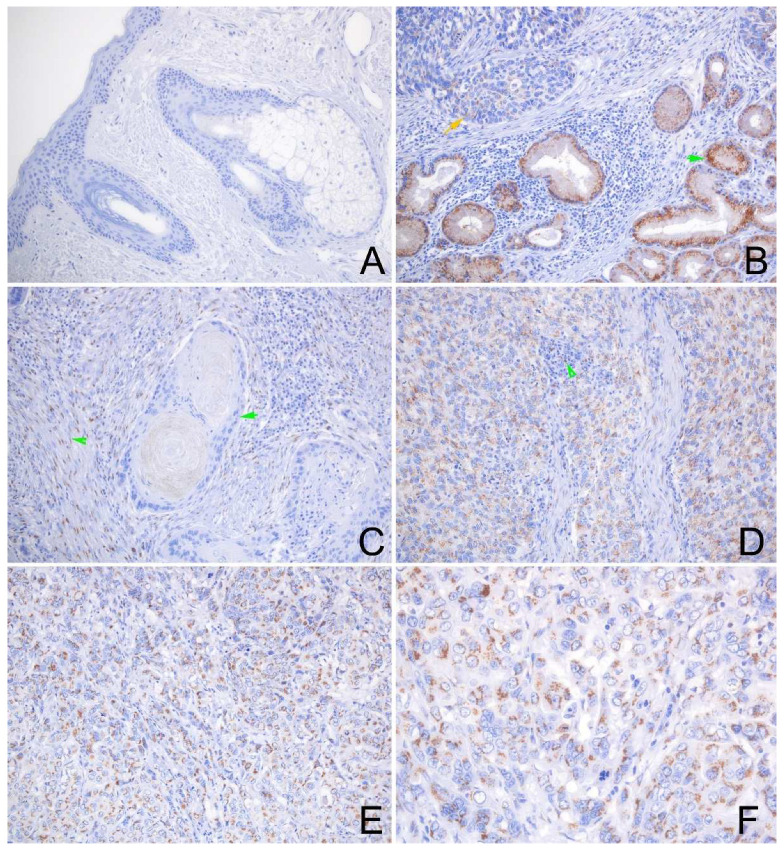
Representative photomicrographs of GOLM1 expression in the normal oral epithelium and HNSCC tissues. (**A**) Shows negative expression of GOLM1 protein in the normal oral squamous epithelium, hair follicle, and sebaceous glands. (**B**) Shows strong positive staining of GOLM1 protein in the normal gland (green arrow), and weak staining of GOLM1 in the SCC (orange arrow). (**C**) Shows negative staining of GOLM1 protein in the well-differentiated SCC, and positive staining of GOLM1 protein in the fibroblast cells. (**D**) Shows positive staining of GOLM1 protein in the tumor cells of moderately differentiated SCC and immune cells (green arrowhead); (**E**,**F**) Shows positive staining of GOLM1 protein in the tumor cells of poorly differentiated SCC. ((**A**–**E**), original magnification ×200; (**F**), original magnification ×400).

**Figure 5 ijms-23-15433-f005:**
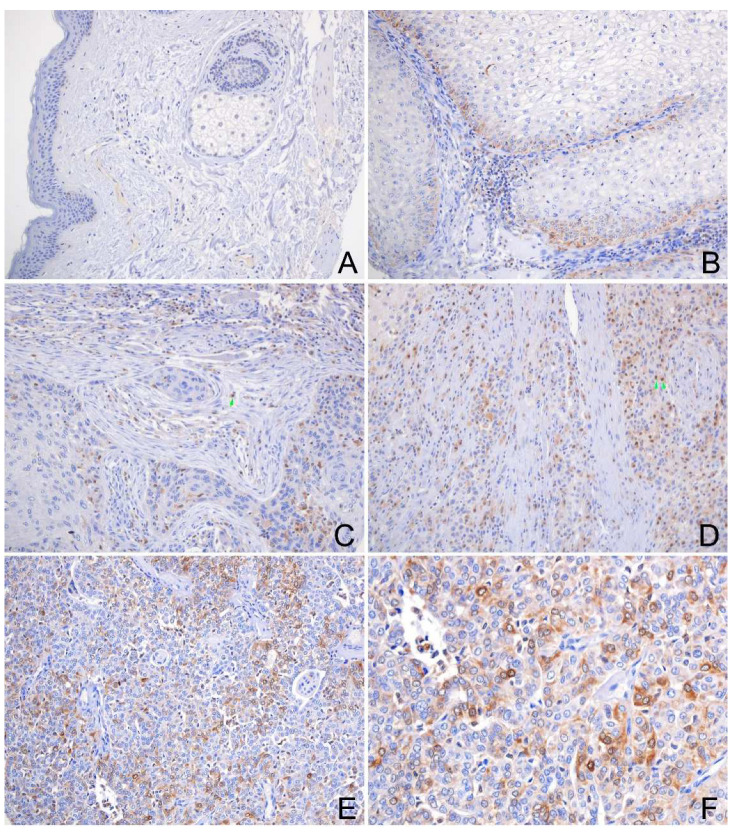
Representative photomicrographs of FAM49B expression in the normal oral epithelium and HNSCC tissues. (**A**) Shows negative expression of FAM49B protein in the normal oral squamous epithelium, hair follicle, and sebaceous glands; (**B**) Shows positive staining of FAM49B protein in the tumor cells around the nest of well-differentiated SCC; (**C**) Shows focal staining of FAM49B protein in the moderately differentiated SCC and immune cells (green arrow). (**D**) Shows positive staining of FAM49B protein in the immune cells of SCC (green arrow), which were localized in the cytoplasm and nuclear; (**E**,**F**) Shows positive staining of FAM49B protein in the tumor cells of poorly differentiated SCC. ((**A**–**E**), original magnification ×200; (**F**), ×400).

**Figure 6 ijms-23-15433-f006:**
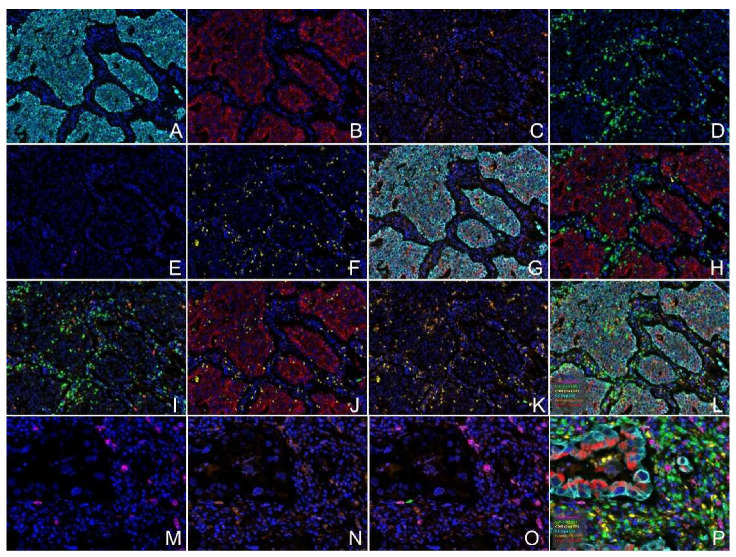
CK, GOLM1, FAM49B, CD3, CD20, and CD68 proteins were detected by mIF in the HNSCC tissues. (**A**): CK; (**B**): GOLM1; (**C**): FAM49B; (**D**): CD3; (**E**): CD20; (**F**): CD68; (**G**): Co-expression of CK, GOLM1, and FAM49B (green arrow showed). (**H**): Co-expression of GOLM1 and CD3 (yellow arrow). (**I**): Co-expression of FAM49B and CD3 (red arrow). (**J**): Co-expression of GOLM1 and CD68 (green arrow), the red signal is VISTA, and the pink signal is CD68. (**K**): Co-expression of FAM49B and CD68 (green arrow). (**L**): Merged images of all markers. (**M**): CD3; (**N**): FAM49B; (**O**): Co-expression of FAM49B and CD20 (green arrow). (**P**): Merged images of all markers. (Original magnification (**A**–**L**) ×200, (**M**–**P**) ×400).

**Table 1 ijms-23-15433-t001:** Top 10 pathways enriched by GSEA analysis in GOLM1 and FAM49B.

	Name of Pathway	NES	NOM *p*-Val	FDR q-Val
GOLM1	Cell adhesion molecules (CAMs)	2.19	0.000	0.003
	Complement and coagulation cascades	2.12	0.000	0.008
	TGF-beta signaling pathway	2.10	0.000	0.007
	Focal adhesion	2.06	0.002	0.011
	Vascular smooth muscle contraction	2.06	0.000	0.009
	ECM–receptor interaction	2.05	0.000	0.010
	Colorectal cancer	2.05	0.002	0.009
	Systemic lupus erythematosus	2.01	0.004	0.015
	Pathways in cancer	1.99	0.000	0.016
	N-Glycan biosynthesis	1.96	0.004	0.020
FAM49B	Antigen processing and presentation	2.33	0.000	0.000
	Oocyte meiosis	2.31	0.000	0.000
	RNA degradation	2.29	0.000	0.000
	NOD-like receptor signaling pathway	2.28	0.000	0.000
	Cytosolic DNA-sensing pathway	2.25	0.000	0.000
	Ubiquitin mediated proteolysis	2.20	0.000	0.001
	RIG-I-like receptor signaling pathway	2.16	0.000	0.002
	Basal transcription factors	2.13	0.000	0.003
	Proteasome	2.12	0.002	0.003
	Pyrimidine metabolism	2.12	0.000	0.003

**Table 2 ijms-23-15433-t002:** High expression of GOLM1 protein was associated with clinicopathological parameters of HNSCC tissues.

Parameters	Cases (*n* = 81)	GOLM1 in Tumor Cells		
		High	Low	*p* Value
Gender				1.000 *
Male	73	10	63	
Female	8	1	7	
Age				0.965 *
≤64	63	8	55	
>64	18	3	15	
Depth of invasion (T)				1.000 *
T1, T2	45	6	39	
T3, T4	36	5	31	
Lymph-node metastasis (N)				0.648
N0	42	5	37	
N1, N2	39	6	33	
Grade				<0.001 *
I, II	58	2	56	
III	23	9	14	
TNM stage				0.272
I + II	24	1	23	
III	41	7	34	
IVA	16	3	13	

* Continuity correction *p* value.

## Data Availability

Publicly available datasets were analyzed in this study. This data can be found here: https://portal.gdc.cancer.gov/, https://www.ncbi.nlm.nih.gov/geo/, http://gepia2.cancer-pku.cn/, accessed on 21 January 2021.

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
