# Peer review of "GOLM1 and FAM49B: Potential Biomarkers in HNSCC Based on Bioinformatics and Immunohistochemical Analysis"

_ijms, 2022, doi:10.3390/ijms232315433_

Round 1
Reviewer 1 Report
The authors mentioned the GOLM1 and FAM49B genes can be used as potential biomarkers to determine the development of HNSCC or predict the prognosis of HNSCC. From the results I don't think the author has sufficient evidence to draw this conclusion. Furthermore, the supplementary information provided by the authors appears to be incorrect.
Author Response
Thank your comments and advices, we have revised our manuscript carefully including the supplementary information. Our study performed the bioinformatics, IHC and mIF to evaluate the significances of GOLM1 and FAM49B genes in HNSCC. Our bioinformatical results demonstrated that high expression of GOLM1 and FAM49B were both significantly correlated with worse 10-year overall survival. Our IHC and mIF results showed expression of GOLM1 and FAM49B proteins was positive in the tumour cells of HNSCC, and negative in the oral squamous epithelium, so we speculated GOLM1 and FAM49B genes can be used as potential biomarkers to determine the development of HNSCC.
Reviewer 2 Report
Dear Author,
My comments on the manuscript entitled «GOLM1 and FAM49B: potential biomarkers in HNSCC based 2 on bioinformatics and immunohistochemical analysis» (ijms-1978048), Yue Xi and co-authors:
1. In my opinion, the beginning in the Introduction section (lines 30-37) needs to be corrected. Why do the authors describe different localizations in HNSCC? Reference 1 can be replaced with reference 9, which describes the pathogenesis of HNSCC.
2. Reference 3: why is it written in Introduction section?
3. Reference 6: in my opinion, it is not needed.
4. Figure 1(F): there is no definition of "AJCC4 stage" in the figure legend. Also "pathologic stages" - please explain. The figure shows only one tumor localization: OSCC. Only tumors of this localization were used in the study?
5. Figure 3 (C) and (D): the authors used https://string-db.org/cgi/network?taskId=bt6XBJTlHci2&sessionId=bxgBZayZocKX. Wouldn't it be better to have a "Predicted Functional Partners" table with proteins and "Score" values, which is located there?
6. Line 171, table 1. In the title, please indicate which pathways - Regulation of signaling pathways? Why "Colorectal cancer" was separately included in the table?
7. Line 289-290: "However, in further studies on tumor aggressiveness and proliferation, different mechanisms mediated by FAM49B have been found in different cancers" - what mechanisms did the authors have in mind? In the results of the study, there is no study of the mechanisms of participation of FAM49B in carcinogenesis in HNSCC.
8. Line 305. Reference 20: For what purpose do the authors cite studies of GOLM1 in the pathogenesis of hepatitis of viral and non-viral etiology?
9. Cytokeratin cocktail AE1 & AE3: Is it a specific marker of HNSCC? Why didn’t the authors use Anti-Smooth Muscle Actin (α-SMA)? How were cells with nuclei distinguished? In my opinion, it is necessary to add explanations to the figures.
10. Discussion section: «We found positive staining for FAM49B protein in the immune cells around SCC and tumour cells of poorly differentiated SCC». Did the authors find the FAM49B protein in tumor-associated immune cells and in tumor cells? Please explain this using for example GOLM1 https://doi.org/10.1038/s41392-021-00673-6.
11. Line 331: please specify which of your analyses showed correlations of GOLM1 and AM49B with immune cells?
12. Line 340: incorrect use of reference 30.

Author Response
Thank your comments and many good advices. We updated our rebuttal letter, please check it, and you have any question about our manuscript, please contact me.

Reviewer 3 Report
The authors Xi et al. conducted research on tumor genetics of HNSCC, using both 44 samples of GSE83519 from Gene Expression Omnibus (GEO) datasets and 546 samples of HNSCC from The Cancer Genome Atlas (TCGA). Afterwards, immunohistochemistry and multiplex immunofluorescence techniques were applied to assess the expression of two genes/gene products and spatial distribution of the gene expression in the HNSCC tumor tissue.
The authors show that GOLM1 and FAM49B genes were associated with worse prognosis. Furthermore, they show that these genes or their respective products are highly expressed in HNSCC tumor cells and associated with a higher disease grading.
The results are very interesting and well discussed. The presentation of the results is appealing and descriptive. The introduction and methods' section gives sufficient background.
Though the most interesting parts are based on publicly available data sets, the results of the manuscript are overall interesting. Regarding the own data from immunohistochemistry a more quantitative descriptive analysis (e.g. IHC H scores) would be able to strengthen the manuscript.
The language is fine, nevertheless, a thorough revision of the text is needed to add missing spaces throughout the text, e.g. in line 37.
Author Response
Thank your positive comments! We read and revise our manuscript carefully, according to the reviewers comments. And we upload our revised manuscript, you can check it again.
Round 2
Reviewer 2 Report
The article has been revised according to my comments.